# Characterization of Three Variants of SARS-CoV-2 In Vivo Shows Host-Dependent Pathogenicity in Hamsters, While Not in K18-hACE2 Mice

**DOI:** 10.3390/v14112584

**Published:** 2022-11-21

**Authors:** Gabriela Toomer, Whitney Burns, Liliana Garcia, Gerelyn Henry, Anthony Biancofiori, Albert George, Ciera Duffy, Justin Chu, Morgan Sides, Melissa Muñoz, Kelly Garcia, Anya Nikolai-Yogerst, Xinjian Peng, Landon Westfall, Robert Baker

**Affiliations:** 1Division of Microbiology and Molecular Biology, Illinois Institute of Technology Research Institute (IITRI), Chicago, IL 60616, USA; 2Department of Pathology, Charles River Laboratories, Inc., Chicago, IL 60077, USA

**Keywords:** SARS-CoV-2, COVID-19, animal models, Omicron B.1.1.529, Omicron BA5.2, Delta B.1.617.2, WA1/2020, LD_50_ survival, K18-hACE2 mice, Syrian hamster, memory T cells, cytokines, T cells, activation-induced memory (AIM)

## Abstract

Animal models are used in preclinical trials to test vaccines, antivirals, monoclonal antibodies, and immunomodulatory drug therapies against SARS-CoV-2. However, these drugs often do not produce equivalent results in human clinical trials. Here, we show how different animal models infected with some of the most clinically relevant SARS-CoV-2 variants, WA1/2020, B.1.617.2/Delta, B.1.1.529/Omicron, and BA5.2/Omicron, have independent outcomes. We show that in K18-hACE2 mice, B.1.617.2 is more pathogenic, followed by WA1, while B.1.1.529 showed an absence of clinical signs. Only B.1.1.529 was able to infect C57BL/6J mice, which lack the human ACE2 receptor. B.1.1.529-infected C57BL/6J mice had different T cell profiles compared to infected K18-hACE2 mice, while viral shedding profiles and viral titers in lungs were similar between the K18-hACE2 and the C57BL/6J mice. These data suggest B.1.1.529 virus adaptation to a new host and shows that asymptomatic carriers can accumulate and shed virus. Next, we show how B.1.617.2, WA1 and BA5.2/Omicron have similar viral replication kinetics, pathogenicity, and viral shedding profiles in hamsters, demonstrating that the increased pathogenicity of B.1.617.2 observed in mice is host-dependent. Overall, these findings suggest that small animal models are useful to parallel human clinical data, but the experimental design places an important role in interpreting the data. Importance: There is a need to investigate SARS-CoV-2 variant phenotypes in different animal models due to the lack of reproducible outcomes when translating experiments to the human population. Our findings highlight the correlation of clinically relevant SARS-CoV-2 variants in animal models with human infections. Experimental design and understanding of correct animal models are essential to interpreting data to develop antivirals, vaccines, and other therapeutic compounds against COVID-19.

## 1. Introduction

Worldwide transmission of the novel severe acute respiratory syndrome coronavirus 2 (SARS-CoV-2) in the human population has contributed to the persistence of the COVID-19 pandemic [1]. The emergence of new variants refers to viral genome adaptation in which mutations in the spike gene RBD (receptor binding domain) drastically increase binding affinity in the RBD–human angiotensin-converting enzyme 2 (hACE2) complex, while also transmitting rapidly in human populations [2,3]. The emergence of new variants in dominantly immunologically naïve populations suggests that adaptive mutations in the viral genome continue to improve viral fitness in unvaccinated and vaccinated individuals. A single amino acid change in the spike protein at position 614 (D614G) in the ancestral strain (i.e., WA1/2020) from aspartic acid to glycine was identified in a small fraction of sequenced samples and quickly became the predominant variant worldwide [4,5]. The fitness advantage conferred by this single amino acid change was supported by major increases in infectivity, viral load, and transmissibility in vitro, as well as in animal models [4].

SARS-CoV-2 variants are still emerging around the world, posing new public health threats due to the differential efficacy of vaccines against multiple SARS-CoV-2 strains. Even in highly endemic regions, new variants have replaced the formerly dominant strains, resulting in infection and mortality spikes [2,3,6].

Among the reported sublineages, B.1.617.1 is designated as a variant of interest and B.1.617.2/ Delta as a variant of concern (VOC) by the World Health Organization. The rise in COVID-19 cases worldwide during the second wave of infection was speculated to be due to the high-transmission potential of the Delta variant. This variant replaced the others in circulation, and was reported in 142 countries. The characteristic mutations reported in the spike gene of the B.1.617 lineage are D111D, L452R, D614G, P618R, and E484Q [4,7]. These mutations suggest increased viral binding to ACE2, transmissibility, secondary attack rate, hospitalization risk, and escape of immune neutralization. The potential impacts of the Delta variant on vaccine and therapeutic effectiveness are uncertain, as limited data are available. Recent studies have reported decreased neutralization efficiency in vaccinated individuals and resistance to monoclonal antibody therapy against the Delta variant [8,9].

A number of Omicron sublineages have been described. The initial Omicron wave was caused by the B.1.1.529 (BA.1) strain which, compared with ancestral WA1/2020, contains 30 amino acid substitutions, 6 amino acid deletions, and 3 amino acid insertions, which are largely clustered at the sites of interaction of potently neutralizing antibodies on the ACE2-interacting surface, around the N343 glycan, and in the NTD [10].

Animal models are required in SARS-CoV-2 preclinical research. Although in vitro, ex vivo, and organoid models are key to reveal some molecular mechanisms of SARS-CoV-2 infection, animal models recapitulate the clinical and pathological characteristics of COVID-19 in humans are required to study viral pathogenesis, transmission, evasion strategies, disease etiology, host responses, therapeutic agents, and vaccines. Several animal models have been used during this pandemic [11]; Althought, the literature is overwhelming regarding rodent models due to their low cost, convenient husbandry requirements, and ease of availability [3,12,13,14,15,16,17], the drawbacks of using mouse models for human viruses are species tropism, species specificity, and immune-response factors. Compared with other lab animals, mice offer many practical advantages, including small sizes, multiple well-established strains, clear genetic background, highly available research tools, and ease of genetic manipulation. Due to the low affinity of the mouse ACE2 (mACE2) for the S protein of SARS-CoV-2, mice cannot be efficiently infected with the ancestral wild-type SARS-CoV-2 variants. The human keratin 18 promoter directs human ACE2 expression to epithelia, specifically the airway epithelia, where the viral infection typically begins, making K18-hACE2 susceptible to SARS-CoV-2, therefore useful for studying antiviral therapies against COVID-19 [3,13,14,15,16,17,18,19]. Syrian hamsters (*Mesocricetus auratus*) rapidly developed into a popular model, as they naturally express ACE2 residues that recognize the SARS-CoV-2 spike protein, making them susceptible to SARS-CoV-2 infection and recapitulating many characteristic features as seen in patients with a moderate, self-limiting course of the disease such as specific patterns of respiratory tract inflammation, vascular endothelialitis, and age dependence mimicking transmission and different courses of the wide spectrum of COVID-19 manifestations in humans [12,13,20,21,22,23].

Immunological memory and response to SARS-CoV-2 infection has been shown to reduce the severity and aid in the protection against infection [24,25]. It is important to understand how the immune system in different animal models responds to infection when translating animal research to humans. CD4+ and CD8+ memory T cells have been found to be induced after infection and vaccination [26,27,28]. These T cells have also been found to protect animals against infection even in the absence of SARS-CoV-2-specific antibodies [1]. Further, CD4+ T cells are critical to aid in the development of antigen-specific memory B cells [1,29]. Thus, understanding how different variants stimulate immunological memory is critical. Of interest, the Omicron variant is known to be able to infect hosts regardless of the ACE2 status [19]. In this report, we investigate the immunological response to Omicron infection in two mouse strains: one which includes the human ACE-2 receptor (K18-hACE2), and the strain of origin (C57BL/6J) which lacks this receptor.

There is now growing evidence that Omicron causes a less severe pathology than the ancestral strain and other VOCs [13,30]. It has been shown that alpha, beta, gamma, and Delta VOCs replicate efficiently in the lungs of Syrian hamsters and to a similar level as the ancestral strain [20,22,23]. Here, we show the infectivity of the Omicron and Delta variants versus the WA1/2020 ancestral strain D614 in mouse and hamster models.

## 2. Results

### 2.1. Enhanced Lethality in B.1.617.2-Infected K18 Mice and Reduced Pathogenicity in B.1.1.529-Infected K18 Mice

We sought to investigate the infective dose (ID_50_) and/or lethal dose (LD_50_) of the three most clinically relevant variants of SARS-CoV-2: B.1.617.2/Delta, the first and most prevalent in the human population in 2021 (known for being highly pathogenic in humans); B.1.1.529/Omicron (BA.1), the variant that displaced Delta and became the most prevalent in 2022 (less pathogenic but more transmissible [19]); and Ancestral WA1/2020, no longer in the population but is the strain that currently approved vaccines are based on, as well as used to test therapeutics against COVID-19. We conducted mouse infection experiments using the ancestral WA1/2020 isolate (containing ancestral D614), B.1.617.2 (also known as 21A Delta or 21A/S:478K with the D614G substitution), and B.1.1.529 (also known as 21K Omicron with D614G and 35 more mutations). The hACE2 transgenic mice (K18-hACE2) (Figure 1A) were used in three independent experiments with several serial dilutions of our viral stocks. Here, we summarize the LD_50_ for WA1/2020 compared to the lower dose of B.1.617.2/Delta and the higher dose of B.1.1.529/Omicron. Mortality started at days 8–9 (8–9 days postinfection (d.p.i.)), where 50% of the animals infected with 1 × 10^3^ TCID_50_ WA1/animal succumbed to infection. Animals infected with a dose of 1 × 10^3^ TCID_50_ Delta/animal presented 100% mortality by day 8 (Appendix A). Interestingly, the lower dose of 1 × 10^1^ TCID_50_ Delta/animal presented 100% mortality by day 11. Conversely, the higher dose (our straight viral stock) of 9 × 10^6^ TCID_50_ Omicron/animal presented no SARS-CoV-2 disease-associated mortality, as shown by the Kaplan–Meier survival analysis (Figure 1B). The body weight loss of the 1 × 10^1^ TCID_50_/Delta-infected mice was significantly greater (*p* = 0.0371) than that of the 9 × 10^6^ TCID_50_/Omicron-infected mice and similar to the 1 × 10^3^ TCID_50_/WA1 across 14 dpi, on average (Figure 1C).

Viral burden in lungs was measured by a TCID_50_ infectivity assay at 3 dpi with 1 × 10^3^ TCID_50_/WA1 (*n* = 7), 1 × 10^1^ TCID_50_/B.1.617.2/Delta (*n* = 7), and 9 × 10^6^ TCID_50_/Omicron (*n* = 4) on randomly selected animals after interim sacrifice. The variant with the highest viral burden was WA1, with an average of 7 × 10^6^ TCID_50_/mg, followed by Delta with 2.5 × 10^5^ TCID_50_/mg, and Omicron with 7.9 × 10^3^ TCID_50_/mg. However, due to high variability and small sample size, no statistically significant differences were found regarding viral burden after the ANOVA analysis (Figure 1D). Therefore, we were unable correlate higher titers and pathogenicity. Furthermore, the animals infected with 1 × 10^1^ TCID_50_ of B.1.617.2/Delta that met the predetermined criteria for moribundity on day 6 were euthanized to avoid unnecessary pain and distress. Lungs were collected from these moribund animals for histopathological evaluation. Moribund lungs were observed to have multifocal necrosis, neutrophilic infiltrates, hemorrhage, edema, loss of pneumocytes, epithelial hyperplasia, infiltration with neutrophils, pulmonary intra-alveolar macrophages (PIMs), and red blood cells when compared to the lungs of the mock-infected animals (Figure 1E).

### 2.2. Host-Dependent Pathogenicity of B.1.617.2, WA1/2020, and BA.5 Infection in Hamsters

In contrast to other rodents, the Syrian hamsters are naturally susceptible to infection with SARS-CoV-2, allowing for them to be rapidly developed into a common model. The hamster SARS-CoV-2 model recapitulates many characteristic features seen in human patients with a moderate, self-limiting course of the disease, such as specific patterns of respiratory tract inflammation, while not producing the outcome of death [12,20,22]. We aimed to elucidate whether the increased pathogenicity of B.1.617.2 compared to WA1 observed in mouse experiments translates to the nonlethal hamster model. To address this, we intranasally infected hamsters with Delta and WA1 at the mouse-lethal dose of 5 × 10^3^ TCID_50_ and the higher dose of our viral stocks for Omicrons BA.1 and BA.5 of 6.5 × 10^5^ TCID_50_/animal.

We found that the same dose of 5 × 10^3^ TCID_50_ for Delta and WA1 (Figure 2A) produced similar outcomes. The body weight loss was similar for both variants in the hamster model. Animals had a weight loss of 4% on average at 4 dpi (Figure 2B). Additionally, viral replication kinetics in lungs (estimated by RT-qPCR on lung tissues from 3 and 6 dpi for WA1 and 4 dpi for Delta) reflect similar copy numbers in both strains (Figure 2C). In hamsters, we were also able to measure the viral shedding profiles (through estimation of viral copy number by RT-qPCR) in oropharyngeal swabs at different time points after infection. Shedding profiles from hamsters infected with 5 × 10^3^ TCID_50_ of Delta, WA1, or 6.5 × 10^5^ TCID_50_/animal of BA.1 or BA.5 show a peak viral load 2 dpi and a consistent decrease in viral load until 4–5 dpi; no significant differences regarding variants were observed. These data suggest that the observed increase in pathogenicity in mice by Delta is relative to the host and not directly related to the viral variant (Figure 2D). In contrast to mice, hamsters appeared to suffer an acute peak of infection before they recover. Virus shedding was consistent between strains on day 2, the highest viral burden in lungs was measured on day 3, and the highest body weight loss was on day 4, regardless of viral strain (Figure 2).

### 2.3. Reduced Pathogenicity of B.1.1.529/Omicron BA1-Infected Mice

We next assessed the infectious dose (ID_50_) for B.1.1.529/Omicron in C57BL/6J mice to investigate the strain’s reported adaptation (due to multiple genetic changes) to no longer require human ACE2 for infectivity [13]. We conducted mouse infection experiments in K18-hACE2 transgenic mice and the nontransgenic strain of origin, C57BL/6J (Figure 3A). Serial dilutions of B.1.1529/Omicron viral stock, from 9 × 10^6^, 1 × 10^5^, 1 × 10^4^, to 1 × 10^3^ TCID_50_/animal in K18-hACE2 mouse, and 9 × 10^6^, 1 × 10^5^, to 1 × 10^4^ TCID_50_/animal in C57BL/6J, were utilized. The K18-hACE2 transgenic mice inoculated with the higher dose of 9 × 10^6^ TCID_50_ was the only group that presented a slight reduction in body weight, with a peak of −1.4% body weight loss at day 7, but on day 8 the animals recovered, and further weight was gained by day 12 (Figure 3B). However, none of the other dosing groups (either K18-hACE2 or C57BL/6J) presented a consistent body weight loss (Figure 3C) or had any other clinical signs. We next compared the viral burden in the lungs by RT-qPCR (Figure 3D) and TCID_50_. The K18-hACE2 mice sustained moderate levels of infection, as shown for the average of 1 × 10^4^ TCID_50_/mg in the lungs (Figure 3E). The levels of viral RNA were comparable to the TCID_50_ assay (Figure 3E,D). Similar results were observed for the C57BL/6J mice infected with 9 × 10^6^ TCID_50_. This group also presented similar levels of viral burden in lungs to K18-hACE2 mice. In this model, we aimed to have an additional readout due to a lack of clinical signs, body weight change, or mortality. For ethical reasons, we were not able to perform nasal washes in mice to obtain viral shedding profiles. However, we performed oropharyngeal swabs in alert animals from days 2 to 4 postinfection. The shedding profiles obtained from the oral swabs show that viral RNA had similar levels (on average) in the K18-hACE2 and C57BL/6J mice infected with 9 × 10^6^ TCID_50_ across time points (*p* > 0.005, multiple regression) (Figure 3F).

### 2.4. Spike-Specific T Cell Phenotypes in Omicron-Infected K18-hACE2 and C57BL/6J Mice

Finally, we assessed the inflammatory responses in the spleens of K18-hACE2 mice at 21 dpi. SARS-CoV-2 spike-specific T cell responses were first measured using flow cytometry activation-induced marker (AIM) assay (Figure 4A and Appendix A) by stimulating cells with an overlapping peptide pool of Omicron’s spike domain [31,32,33]. When comparing AIM+ phenotypes, there was no difference in the percentage of CD8+ AIM+ cells, regardless of the mouse model or Omicron challenge status. In contrast, in K18-hACE2 mice, the baseline of uninfected CD4+ AIM+ T cells was significantly lower than in infected K18-hACE2 mice (inner left, Figure 4A). This was not observed in C57BL/6J CD4+ T cells. However, when unstimulated controls were subtracted from the Omicron-spike-peptide-treated samples, this effect was abolished (Figure 4A). This phenomenon displays that there was not a marked observed difference between antigen-stimulated and unstimulated wells, regardless of challenge. It may also demonstrate that baseline antigen specificity was altered by the presence of the challenge in K18-hACE2 mice that was not present in the C57BL/6J mice.

SARS-CoV-2 memory T cells were measured by stimulating cells with an overlapping peptide pool of Omicron’s spike domain. Four T cell memory phenotypes were considered via flow cytometry: naïve (Tn; CD45RA+ CCR7+), central memory (Tcm; CD45RA− CCR7+), effector memory (Tem; CD45RA− CCR7−), and terminally differentiated effector memory (Temra; CD45RA+ CCR7−) cells. Regardless of AIM status, there was no significant difference in T cell memory phenotype in the CD8 T cells between the infected and uninfected animals (Figure 4B and Figure 5A, left). In contrast, both CD4+ T cells and CD4+ AIM+ T cells in K18-hACE2 infected mice observed a significant decrease in Tcm cells and an increase in Tem cells in comparison to uninfected K18-hACE2 mice (Figure 4B and Figure 5A, right). Interestingly, CD4+ AIM+ cells were mostly Tem cells, with Tcm second (Figure 4B, right). In contrast, CD4+ T cells without sorting for AIM+ were mostly Tcm cells, with Tem second (Figure 5A, right). Thus, while the trend of increase/decrease in the presence of Omicron infection remained the same, the overall pool of memory T cells shifted when observing activation-induced cells towards Tem.

Next, SARS-CoV-2-specific T cell responses were measured by intracellular cytokine staining (ICS) for IFN-γ, TNF-α, or Granzyme B. Interestingly, while CD4+ T cells showed a possible increase in AIM+ cells and a change in memory phenotype, we did not observe significant differences in cytokine production. Instead, cytokine production differences were observed in CD8+ T cells in K18-hACE2 mice (Figure 4C and Figure 5B). No significant increases in cytokine production were observed in C57BL/6 mice (data not shown). Both CD8+ and AIM+ CD8+ T cells expressed a significant increase in IFN-γ, TNF-α, and IFN-γ+ TNF-α+. A significant increase was not observed with Granzyme B or triple producers. Data were further analyzed to assess the distribution of single, double, and triple cytokine-producing T cells in K18-hACE2 mice. Figure 4D and Figure 5C show that without Omicron infection, most cytokine-producing CD8 T cells only produce one cytokine. In contrast, mice that were challenged with Omicron were dominantly double producers.

## 3. Discussion

In COVID-19 patients with acute respiratory illness, the main clinical manifestation is pneumonia [4]. Consistent with previous studies, our data suggest that WA1/2020, B.1.617.2/Delta, and B.1.1.529/Omicron SARS-CoV-2 variants replicate efficiently in the lungs of the K18-hACE2 heterozygous mouse and Syrian hamster models [4]. Our results indicate that, in mice, Delta is most pathogenic, followed by WA1, and Omicron was absent of clinical signs. This is similar to what has been described for human infections population-wise [4]. Interestingly, Omicron was able to infect C57BL/6J mice that do not have the human ACE2 receptor. Viral shedding profiles and viral titers in the lungs were similar between the K18-hACE2 and the C57BL/6J. These data indicate viral adaptation to a new host and a confirmation of asymptomatic SARS-CoV-2 carriers that accumulate and shed the virus even in the absence of clinical signs.

Next, we show how the same dose of Delta and WA1 have similar viral replication kinetics, pathogenicity, and viral shedding profiles in hamsters. This demonstrates that the increased pathogenicity observed in mice is host-dependent and not associated with viral replication.

In contrast to similar viral shedding profiles and titers between K18-hACE2 and C57BL/6J mice, our data suggest that K18-hACE2 mice, but not C57BL/6J mice, have a change in the CD4 memory T cell repertoire towards effector memory cells and an increase in CD8 T cell cytokine production after Omicron infection. Since K18-hACE2 mice demonstrated changes in the T cell immune response, while C57BL/6J mice did not, this suggests that the hACE2 receptor is necessary for generating immunological memory and T cell cytokine production against SARS-CoV-2.

While we did not observe significant changes in the CD8 T cell memory markers, only cytokine production, this may be due to the timing of splenocyte collection. Typically, CD8 memory T cells spike 8 dpi after the initial encounter with a pathogen and reduce to 5–10% of that initial population over the following weeks [34]. Due to this natural reduction over time, 21 dpi may have been enough time for these memory T cells to return near to baseline levels. Of note, we observed a similar trend to the CD4 memory cells (an increase in Tem) and a slight decrease in Temra, but these changes were not significant. We do observe antigen-specific cytokine responses, which does suggest that memory CD8 T cells were generated from the initial challenge and restimulated with the Omicron spike peptides. Further studies should be performed using peptides from other regions of Omicron (such as the nucleocapsid) and at multiple time points to determine the full T cell response against the virus in K18-hACE2 mice, as it has been reported that memory CD4 T cells can bind the M, spike, N, nsp3, nsp4, ORF3a, and ORF8 regions (among others), while CD8 can also bind the M region and multiple ORFs [35]. Overall, these findings suggest that small animal models are useful to parallel human clinical data, but the experimental design places an important role in interpreting the data.

## 4. Materials and Methods

No statistical methods were used to predetermine the sample size. Investigators were not blinded to allocation during experiments and outcome assessment. Randomized animals were assigned to groups based on body weights that will produce similar group mean values by using Ascentos^®^ version 2.0, (PDS Pathology Data Systems, Inc., Basel, Switzerland). No animal body weight varied from the mean of the group body weight by more than 20%.

### 4.1. Ethics Statement

Animal studies were carried out in accordance with the recommendations in the Guide for the Care and Use of Laboratory Animals (National Research Council, 2011) and the Public Health Service Policy on Humane Care and Use of Laboratory Animals (NIH, 2002). The protocols were approved by the Institutional Animal Care and Use Committee at IITRI (Assurance Number D16-00299 A3475-01). Virus inoculations were performed under induced anesthesia and maintained with isoflurane or ketamine hydrochloride and xylazine, and all efforts were made to minimize animal suffering. SARS-CoV-2 in vivo and in vitro studies were performed in the animal biosafety level 3 (ABSL3) facility at IITRI. All experiments complied with all relevant ethical regulations.

### 4.2. Cell Culture

Vero-E6 (Vero E6; ATCC^®^ CRL-1586™) and Vero E6-hACE2-TMPRSS2 (BEI cat. No. NR-54970) cells, which were modified to express higher levels of transmembrane protease, serine 2, and angiotensin-converting enzyme 2, were cultured at 37 °C in Dulbecco’s Modified Eagle medium (DMEM, Gibco, cat. No. 11054001) supplemented with 10% fetal bovine serum (FBS, Gibco, cat. No. A3840001) and 100 U/mL of penicillin–streptomycin (Gibco, cat. No. 15140122). Vero-hACE2-TMPRSS2 cells were supplemented with 10 μg/mL of puromycin (Fisher, cat. No. BP2956100). All cells routinely tested negative for mycoplasma using a PCR-based assay.

### 4.3. Mouse Experiments

Heterozygous K18-hACE2 C57BL/6J mice (strain 2B6.Cg-Tg(K18-ACE2)2Prlmn/J) and C57BL/6J were obtained from The Jackson Laboratory (Bar Harbor, ME, USA). Animals were 14- to 17-week-old female mice and 22–24 g of body weight. Mice were housed in groups and fed standard #2014C Teklad Global 14% Protein Rodent Diet. During challenge, animals were anesthetized with either isoflurane (inhaled) and intranasally challenged with a volume of 30 μL containing: 1 × 10^0^, 1 × 10^2^, 1 × 10^3^, 5 × 10^3^, 1 × 10^4^, and 1 × 10^5^ TCID_50_/animal of WA1/2020; 5 × 10^0^, 5 × 10^1^, 5 × 10^2^, 5 × 10^3^, 5 × 10^4^, and 2.5 × 10^5^ TCID_50_/animal of Delta B.1.617.2; or, 1 × 10^3^, 1 × 10^4^, 1 × 10^5^, and 9 × 10^6^ TCID_50_/animal of Omicron B.1.1.529.

### 4.4. Hamster Experiments

Eight- to twelve-week-old male Syrian golden hamsters of about 90–105 g were obtained from Envigo. Animals were single-housed and fed standard #2014C Teklad Global 14% Protein Rodent Diet. Under ketamine/xylazine (IP) anesthesia, hamsters were intranasally challenged with a volume of 50 μL containing 5 × 10^3^ TCID_50_/animal of WA1/2020 or 5 × 10^3^ TCID_50_/animal of B.1.617.2, or 6.5 × 10^5^ TCID_50_/animal of B.1.1.529 (BA.1) or 6.5 × 10^5^ TCID_50_/animal of BA5.2 SARS-CoV-2 variants. Prior to challenge, hamsters were monitored for 3 days to measure body weight (average of 2 closest days to challenge were used for establish baseline). Body weights were monitored every day for 4 dpi. Oral swabs were collected on 1 to 4 dpi for analyzing viral shedding profiles by RT-qPCR. On 4 dpi, all animals were euthanized. Immediately, the left lung was inflated for histopathological analysis, and the right lung was processed and flash-frozen for RT-qPCR and TCID_50_ assay.

### 4.5. Histopathological Analysis

Fixed tissues were processed through paraffin blocks. Each section from the right cranial, right accessory, right caudal, and right middle lobes were trimmed, embedded in paraffin, sectioned at approximately 5 μm, and stained with hematoxylin and eosin (H&E). All paraffin H&E slides were evaluated microscopically and graded for presence and severity of pathology by a board-certified (ACVP) veterinary pathologist by PAI-Charles River Laboratories, Inc.

### 4.6. Virological Analysis

SARS-CoV-2 isolates were obtained from BEI Resources WA1 or hCoV-19/USA-WA1/2020 (Lineage A.1) NR-52281, Delta or USA/MD-HP05647/2021 (Lineage B.1.617.2) NR-55672, and Omicron hCoV-19/USA/MD-HP20874/2021 (Lineage B.1.1.529), NR-56461. BA.5/Omicron was isolated by IITRI and deposited in BEI and Gene Bank Accession with No. OP654930.1. Stocks were prepared by infection of Vero E6 cells in 2% FBS DMEM for three days and clarified by centrifugation. The titer of the stock was determined through a certified TCID_50_ assay. This seed stock was sequence verified by Illumina Next-Generation Sequencing to confirm the genome stability and avoid introduction of adventitious mutations and certified mycoplasma-free by PCR. All virus experiments were performed in an approved biosafety level 3 (BSL-3) facility at MMB Division of IITRI.


**No.**

**ID**

**Name**

**BEI No.**

**IITRI Lot No.**

**Stock Titer**

**TCID**
^
**50**
^
**/mL**
1WA1hCoV-19/USA-WA1/2020 (Lineage A.1)NR-5228120200924A8.5 × 10^5^2DeltaUSA/MD-HP05647/2021 (Lineage B.1.617.2)NR-55672202110012.5 × 10^6^3Omicron BA.1hCoV-19/USA/MD-HP20874/2021 (Lineage B.1.1.529)NR-56461202202102.9 × 10^8^4Omicron BA.5SARS-CoV-2/human/USA/P2/2022 (BA.5.2)This paper and NRS-58888
202209011.37 × 10^7^

### 4.7. Oral Swabs

On alert animals, an oropharyngeal (OP) sample was taken using Puritan™ PurFlock™ Ultra Flocked Swabs (Thermo Fisher Scientific cat. No. 22-025-192). Swabs were vortexed for 10 s in 300 µL of 1× DNA/RNA Shield^TM^ (Zymo Research, cat. No. R1100) and centrifuged for 5 min at 5000× *g*. Samples were extracted using Quick-RNA Viral Kit (Zymo Research cat. No. R2141) and the viral genome copy numbers were evaluated by RT-qPCR analysis.

### 4.8. Nasal Washes

Hamsters were anesthetized with ketamine (100 mg/kg) and xylazine (5 mg/kg) mixture (KX) via intraperitoneal (IP) injection. Antisedan^®^, an α-2-antagonist, was used to reverse of KX after the procedure. Once the hamster was anesthetized, 0.16 mL of sterile 1X PBS containing penicillin (100 U/mL), streptomycin (100 μg/mL), and gentamicin (50 μg/mL) was injected into the nostrils 20 μL at a time and pipetted out into a collection tube. The recovered nasal wash volume was collected and recorded. Samples were stored at ≤−65 °C until analysis.

### 4.9. TCID_50_ Assay

Lung tissue samples had viral titers determined using the TCID_50_ assay. Briefly, clarified tissue supernatants were diluted 1:10, 1:100, or 1:1000 followed by 2-fold serial dilutions and added to a 96-well plate preseeded with VeroE6 cells. Each sample was plated in triplicate. The plates were read after 72 h for cytopathic effect (CPE) and immunostained.

### 4.10. One-Step RT-qPCR Assay

Following left lung tissue homogenization and brief centrifugation, a sample of 100 µL of the supernatant of the homogenate was mixed with 100 µL of the 2× DNA/RNA Shield^TM^ (1:1 *v*/*v*). The concentration of viral genome copy number in harvested tissue supernatants was determined by RT-qPCR assay. Briefly, RNA was extracted from samples stored in RNA/DNA Shield using the Quick-RNA Viral Kit (Zymo Research) according to the manufacturer’s protocol. The following One-step RT-PCR cycling conditions were used: 50 °C for 15 min (RT), then 95 °C for 2 min (denature), then 40 cycles of 10 s at 95 °C, 45 s at 62 °C.

Primers/probe used for SARS-CoV-2 detection are given below:

2019-nCoV_N1-F: 5′-GACCCCAAAATCAGCGAAAT-3′

2019-nCoV_N1-R: 5′-TCTGGTTACTGCCAGTTGAATCTG -3′

Probe: 2019-nCoV_N1-P: 5′-FAM-ACCCCGCATTACGTTTGGTGGACC-BHQ1-3′

### 4.11. Splenocyte Isolation

Splenocytes were collected from the Omicron B.1.1.529 infected mice 21 days post-infection. A spleen was removed from the animal and collected in media (90% RPMI 1640 (Gibco Cat. 11875093, Waltham, MA, USA) + 10% FBS (VWR Cat. 89510-188, Radnor, PA, USA)). After, the spleen was smashed through a 70 µm mesh strainer (Corning cat. 352350, Corning, NY, USA), and subsequently, red-blood-cell lysed according to manufacturer’s instructions (Invitrogen cat. 00430054, Waltham, MA, USA). Isolated splenocytes were preserved in cryomedia (45% RPMI 1640 + 45% FBS + 10% DMSO (Sigma cat. D2650, St. Louis, MO, USA)), frozen in Mr. Frosty freezing containers (ThermoFisher cat. 5100-0001, Waltham, MA, USA) at ≤−65 °C, then stored in the vapor phase of liquid nitrogen until use.

### 4.12. Flow Cytometry

Splenocytes were cultured for 6 h at 37 °C, 5% CO_2_ in a 96-well U-bottom plate at 1 × 10^6^ cells per well. Culture conditions were as follows: the presence of the Omicron-spike-specific peptides (1.85 µg/mL, JPT Peptide Technologies cat. PM-SARS2-SMUT08-1, Germany), the presence of PMA/ionomycin as a positive control (2 µL/mL; ThermoFisher cat. 00-4970-93), or in the absence of peptides as a negative control. GolgiPlug (1 µL/mL, BD Biosciences cat. 555029, Franklin Lakes, NJ, USA) was added after 2 h to wells that received cytokine staining downstream. Following cell culture, wells were stained with Zombie Viability Dye Aqua (1 µL/mL, BioLegend cat. 423101) for 15 min at room temperature, Fc-blocked with TruStain FcX PLUS (2.5 µg/mL, BioLegend cat. 156603) for 5 min at 2–8 °C, then stained with respective surface antibodies for 30 min at 2–8 °C. The panel for surface memory phenotype analysis consisted of CD3 Alexa Fluor 700 (10 µg/mL, BioLegend cat. 100216), CD4 FITC (0.625 µg/mL, BioLegend cat. 100406), CD8 APC Fire 750 (0.625 µg/mL, BioLegend cat. 100765), CD45RA BV786 (1 µg/mL, BD Biosciences cat. 747759), CD69 BV605 (5 µg/mL, BioLegend cat. 104530), CD134 (OX40) APC (2.5 µg/mL, BioLegend cat. 119414), CD137 (4-1BB) PE (2.5 µg/mL, BioLegend cat. 106106), and CD197 (CCR7) BV421 (5 µg/mL, BioLegend cat. 120120). Wells were fixed with BD Fixation/Permeabilization solution (BD Biosciences cat. 51-2090KZ) for 20 min on ice, then proceeded to flow cytometric analysis after washing and resuspension in FACS buffer. The panel for cytokine phenotype analysis consisted of surface labeling with CD3 Alexa Fluor 700, CD4 FITC, CD8 APC Fire 750, CD69 BV605, CD134 APC, and CD137 PE. Wells were fixed with BD Fixation/Permeabilization solution for 20 min on ice, then stained for 30 min at 2–8 °C with the following cytokine antibodies: Granzyme B (GzB) BV421 (4 µg/mL, BioLegend cat. 396414), IFN-gamma (IFN-γ) PE/Dazzle 594 (0.6 µg/mL, BioLegend cat. 505845), and TNF-alpha (TNF-α) BV785 (1.25 µg/mL, BioLegend cat. 506341). Afterwards, the wells were washed and resuspended in FACS buffer and proceeded to flow cytometric analysis.

Flow cytometry was performed on a BD FACSCelesta (BD Biosciences) using FACSDiva version 9.0 (BD Biosciences). Data were analyzed on FlowJo version 10.5.0 (BD Biosciences). Antigen-specific T cells were measured as a percentage of AIM+ (OX40+ CD137+) CD4+ and (CD69+ CD137+) CD8+ T cells.

The gates applied for identifying AIM+ CD4+ and AIM+ CD8+ T cells producing cytokines were defined according to the cells cultured without stimulation for each individual sample. A Boolean analysis was performed to define the multifunctional profiles on FlowJo. The analysis included IFN-γ, TNF-α, and Granzyme B gated on CD3+ CD4+ AIM+ or CD3+ CD8+ AIM+ cells. The overall response to SARS-CoV-2 was defined as the sum of the background subtracted responses to each combination of individual cytokines. The average positive relative CD4+ and CD8+ T cell responses were calculated per group to define the proportion of multifunctional antigen-specific T cell responses. Statistics for two populations were assessed via Student’s *t*-test and statistics for greater than two populations were assessed via one-way ANOVA on GraphPad Prism 9.

## Figures and Tables

**Figure 1 viruses-14-02584-f001:**
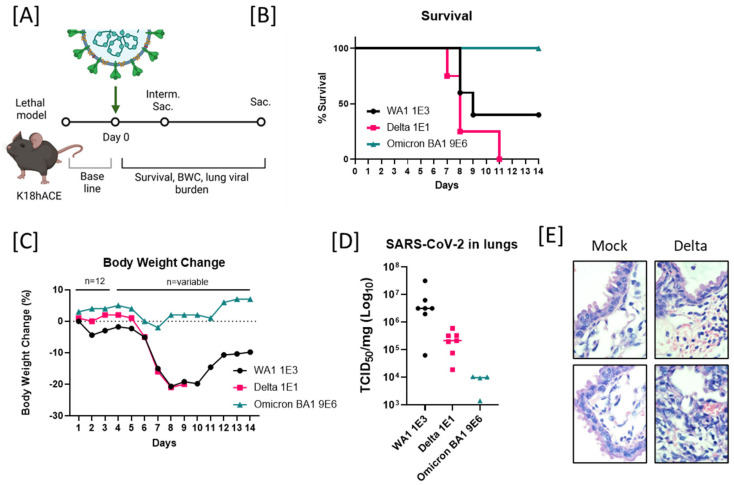
B.1.617.2 (Delta) showed enhanced lethality in infected K18-hACE2 mice. (**A**) Schematic diagram of SARS-CoV-2 infection and experimental design. Interim Sac = scheduled interim sacrifice/euthanasia. Sac. = final euthanasia. Created with BioRender.com. Animals (*n* = 12) were intranasally inoculated with SARS-CoV-2 on day 0. Body weights before challenge were used to establish baseline. Survival and body weight change (BWC) were observed or collected daily. Lungs were collected at the interim and terminal euthanasia. (**B**) Survival in mice model after 14 days postchallenge with 1 × 10^3^ TCID_50_ of WA1, 1 × 10^1^ TCID_50_ of B.1.617.2 (Delta), and 9 × 10^6^ TCID_50_ of B.1.1.529/Omicron BA1. (**C**) Body weight changes comparison between 1 × 10^3^ TCID_50_ of WA1, 1 × 10^1^ TCID_50_ of B.1.617.2 (Delta), and 9 × 10^6^ TCID_50_ of B.1.1.529/Omicron BA1 for 14 days postchallenge. (**D**) Viral titers in lungs of infected mice at 3 days postchallenge with 1 × 10^3^ TCID_50_ of WA1 (*n* = 7), 1 × 10^1^ TCID_50_ of B.1.617.2 (Delta, *n* = 7), and 9 × 10^6^ TCID_50_ of B.1.1.529/Omicron BA1 (*n* = 4). No statistical differences were found (*p* = 0.1553). (**E**) Histopathological analysis with Hematoxylin and Eosin stain (H&E) of the lungs from uninfected and B.1.617.2 (Delta) of SARS-CoV-2-infected mice at 6 days postchallenge under 100× (*n* = 2). TCID_50_ is a quantitation of virus infectivity and stands for the median **T**issue **C**ulture **I**nfectious **D**ose (TCID_50_) defined as the dilution of a virus required to infect **50%** of a given cell culture.

**Figure 2 viruses-14-02584-f002:**
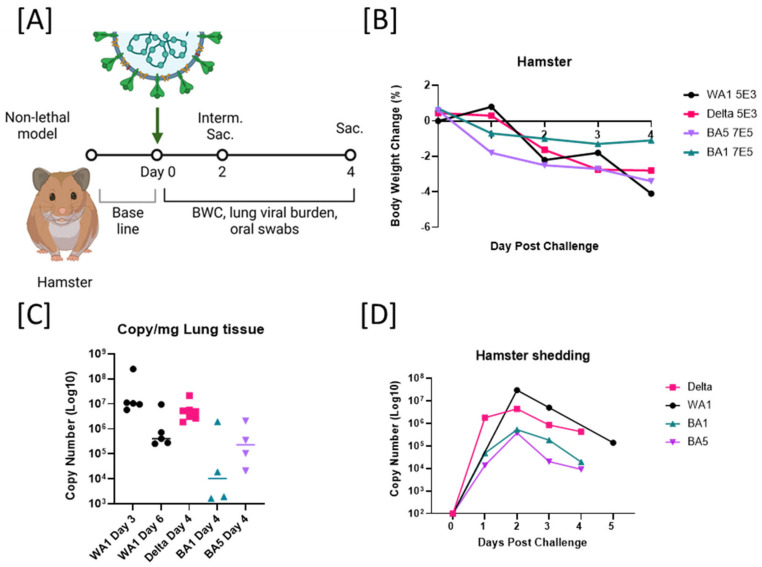
Host-dependent pathogenicity of B.1.617.2 (Delta), WA1/2020 and B1.1.529 (BA.1) and BA.5 Omicron in hamster. (**A**) Schematic diagram of SARS-CoV-2 infection and experimental design. Interim Sac = scheduled interim sacrifice/euthanasia. Sac. = final euthanasia. Created with BioRender.com. Animals were intranasally inoculated with 5 × 10^3^ TCID_50_/animal of SARS-CoV-2 WA1 or B.1.617.2/Delta or 6.5 × 10^5^ TCID_50_/animal of BA.1 or BA.5 on day 0. Body weights before challenge were used to establish baseline. Survival, body weight changes (BWC), oral swabs, or nasal washes were observed or collected daily. Lungs were collected at the interim and terminal euthanasia on days 3 and 6 for WA1 and day 4 for B.1.617.2 (Delta), and BA.1 and BA.5 (Omicron). (**B**) Body weight change comparisons between WA1, B.1.617.2 (Delta), BA.1, and BA.5 for 4 dpi (and 5 dpi for WA1). (**C**) Viral titers estimated as genomic RNA copy number in lungs of infected mice at 3, 4, or 6 days postchallenge with 5 × 10^3^ TCID_50_ of WA1 (*n* = 7), 5 × 10^3^ TCID_50_ of B.1.617.2 (Delta, *n* = 7), or 6.5 × 10^5^ TCID_50_ (*n* = 8) of BA.1 or BA.5 (*n* = 8). (**D**) Viral shedding profiles were estimated by RNA viral copy number in oral swabs collected from day 0 to day 5 dpi. Dpi = days postinfection. TCID_50_ = **50%** of the**T**issue **C**ulture **I**nfectious **D**ose (TCID_50_).

**Figure 3 viruses-14-02584-f003:**
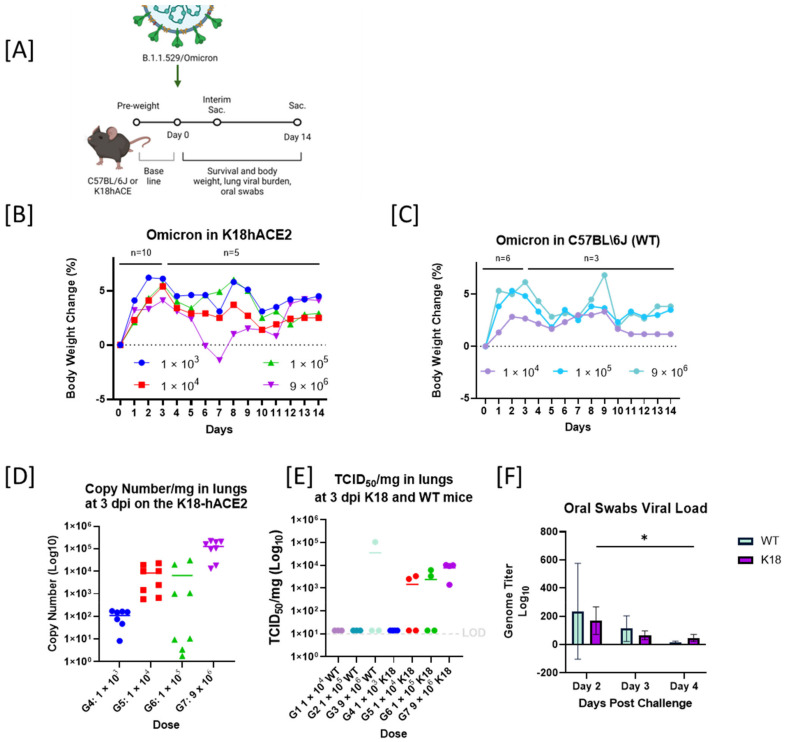
Reduced pathogenicity in B.1.1.529/Omicron (BA.1)-infected mice. (**A**). Schematic diagram of SARS-CoV-2 infection and the experimental design. Created with BioRender.com. Animals were intranasally inoculated with 9 × 10^6^, 1 × 10^5^, 1 × 10^4^, or 1 × 10^3^, in K18-hACE2 mice and 9 × 10^6^, 1 × 10^5^, 1 × 10^4^ in C57BL\6J. of B.1.1.529 variant of SARS-CoV-2 on day 0. Body weights before challenge were used to establish baseline. Survival, body weight, and oral swabs were observed or collected daily. Lungs were collected at the interim and terminal euthanasia. (**B**) Percentage of body weight changes (BWC) in K18-hACE2 mice inoculated with 9 × 10^6^, 1 × 10^5^, 1 × 10^4^, or 1 × 10^3^. All groups started with *n* = 10, *n* = 5 were scheduled for interim euthanasia at 3 dpi. From 4 dpi to 14 dpi, *n* = 5. (**C**) Percentage of BWC in WT mice (C57BL\6J) inoculated with 9 × 10^6^, 1 × 10^5^ or 1 × 10^4^. All groups started with *n* = 6, and *n* = 3 were scheduled for interim euthanasia at 3 dpi. From 4 dpi to 14 dpi, *n* = 3. (**D**) Viral load as RNA genome copy number per mg of lung tissue of K18-hACE2 mice infected with 9 × 10^6^, 1 × 10^5^, 1 × 10^4^, or 1 × 10^3^ of B.1.1.529 SARS-CoV-2 after 3 dpi (*n* = 4 in technical duplicates). (**E**) Viral load by TCID_50_ assay in all WT and K18-hACE2 mice infected with B.1.1.529. (**F**) Virus shedding as viral genome copy number (by RT-qPCR) in oral swabs taken at 2, 3, and 4 dpi of K18-hACE2 and WT mice intranasally inoculated with 9 × 10^6^ TCID_50_/animal (groups 3 and 7 only). Asterisk (*) represent statistical differences by ANOVA *p* ≤ 0.05.

**Figure 4 viruses-14-02584-f004:**
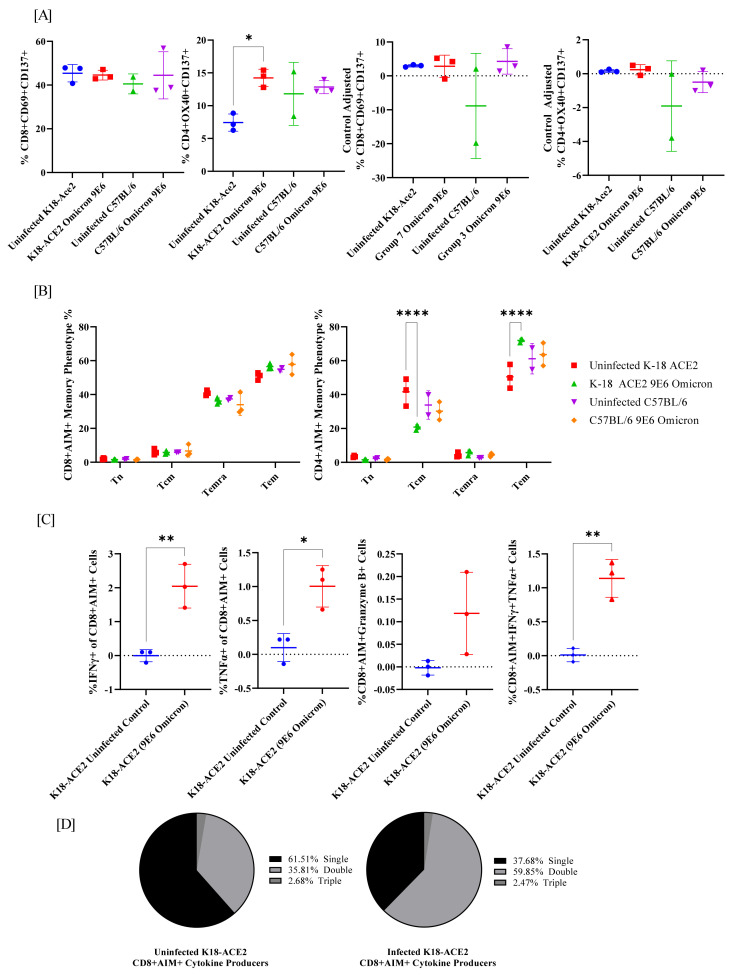
Omicron infection increases CD4+ AIM+ Tem population and promotes cytokine production in AIM+ CD8+ T cells in K18-hACE2 mice. (**A**) CD8 and CD4 AIM+ T cell populations in two mouse models after Omicron challenge. Graphs on the left are Omicron-peptide-stimulated samples unadjusted for control wells, graphs on the right have unstimulated control wells subtracted from Omicron-peptide-stimulated wells. (**B**) CD8+ AIM+ and CD4+ AIM+ memory phenotype was analyzed by gating naïve (Tn; CD45RA+ CCR7+), central memory (Tcm; CD45RA− CCR7+), effector memory (Tem; CD45RA− CCR7−), and terminally differentiated effector memory (Temra; CD45RA+ CCR7−) cells. (**C**) CD8+ AIM+ T cells producing IFN-gamma, TNF-alpha, or Granzyme B by intracellular cytokine staining. (**D**) Multifunctional activity profiles of CD8+ AIM+ T cells evaluated from IFN-gamma, TNF-alpha, and Granzyme B. Asterisks represent statistical significance by ANOVA test where (*) *p* ≤ 0.05; (**) *p* ≤ 0.01; (****) *p* ≤ 0.0001; if (ns) *p* > 0.05 not shown.

**Figure 5 viruses-14-02584-f005:**
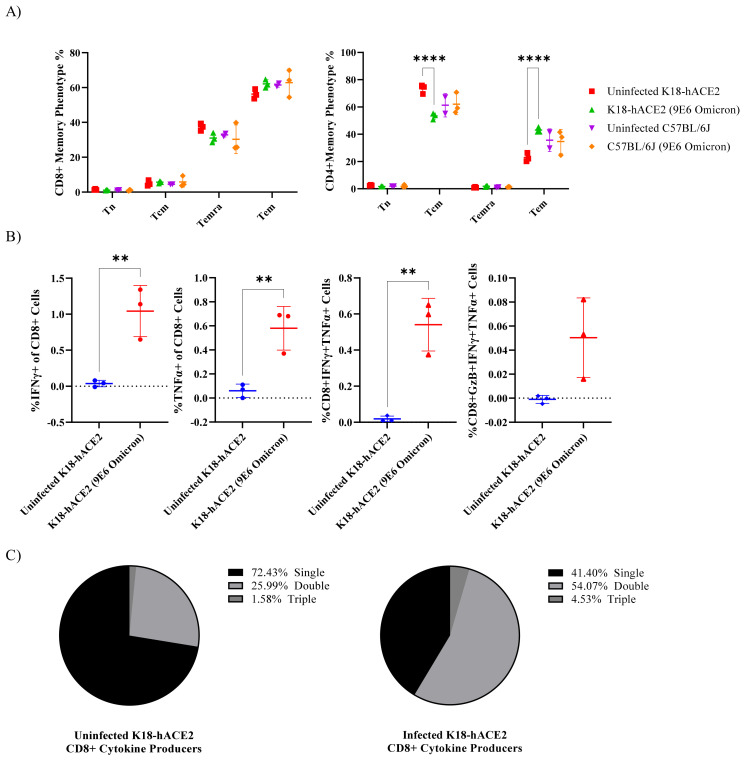
Omicron infection increases CD4+ Tem population and promotes cytokine production in CD8+ T cells in K18-hACE2 mice. (**A**) CD8+ and CD4+ memory phenotype was analyzed by gating naïve (Tn; CD45RA+ CCR7+), central memory (Tcm; CD45RA− CCR7+), effector memory (Tem; CD45RA− CCR7−), and terminally differentiated effector memory (Temra; CD45RA+ CCR7−) cells. (**B**) CD8+ T cells producing IFN-gamma, TNF-alpha, or Granzyme B by intracellular cytokine staining. (**C**) Multifunctional activity profiles of CD8+ T cells evaluated from IFN-gamma, TNF-alpha, and Granzyme B. Asterisks represent statistical significance by ANOVA test where (**) *p* ≤ 0.01; (****) *p* ≤ 0.0001; if (ns) *p* > 0.05 not shown.

## Data Availability

BA.5.2/Omicron was isolated by IITRI under BSL3 and deposited in BEI (NRS-58888) and Gene Bank Accession (OP654930.1).

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
