# Peer review of "Characterization of Three Variants of SARS-CoV-2 In Vivo Shows Host-Dependent Pathogenicity in Hamsters, While Not in K18-hACE2 Mice"

_viruses, 2022, doi:10.3390/v14112584_

Round 1
Reviewer 1 Report
The research presents useful data comparable to human clinical data involving clinically extremely relevant SARS-CoV-2 variants in small animal models. The manuscript is well written.
Author Response
Response to Reviewer 1 Comments
Point 1. The research presents useful data comparable to human clinical data involving clinically extremely relevant SARS-CoV-2 variants in small animal models. The manuscript is well written.
Response 1. We thank reviewer 1 for the comments and agreed that our research in animal models parallels human clinical data, which highlights the importance of rodents as a SARS-CoV-2 model for pre-clinical trials.
Reviewer 2 Report
- Major comments:
In the manuscript, Toomer et al. found that different SARS-CoV-2 variants had different outcomes in mice and hamster models. Specifically, B.1.617.2 is more pathogenic in K18 mice while not in hamsters. B.1.1.529 showed an absence of clinical signs in both mice and hamster models, and it was able to infect wildtype C57BL/6J mice. While similar viral shedding profiles and titers between K18-hACE2 and 306 C57BL/6J mice were observed, B.1.1.529-infected K18-hACE2 mice had different T cell profiles compared to infected C57BL/6J mice.
The study is beneficial in the SARS-CoV-2 experimental design and our understanding of COVID-19 animal models that are helpful in developing antivirals, vaccines, and other therapeutic compounds against COVID-19.
The study can be further improved by including the following suggestions listed in specific comments.
- Specific comments:
1) Title, “Characterization of Three Variants of SARS-CoV-2 in vivo Shows Host-Dependent Pathogenicity in Hamsters”, why were only hamsters mentioned here?
2) Line 22, suggests replacing “ACE2” with “K18-hACE2”.
3) Line 19, suggests replacing “mice” with “K18-hACE2 mice”.
4) Line 21, “B.1.1.529-infected ACE2 mice had different T cell profiles compared to infected K18-hACE2 mice”, here, it should be “B.1.1.529-infected C57BL/6J mice had different T cell profiles compared to infected K18-hACE2 mice”.
5) Introduction part, a brief introduction/summary should be provided about the SARS-CoV-2 animal models of mice and hamsters, and what questions the study will address.
6) Figure 1A, the full names of the abbreviations should be provided in the figure legend.
7) Line 108-109, “Animals infected with a dose of 1 × 103 TCID50 Delta/animal presented 100% mortality by Day 8 (data not shown).” How many animals did the study use? Suggests including this result in supplementary data for the comparison of the same dose of WA1/2020.
8) Line 122, Line 124, and Line 126, the descriptions here “9 × 109 TCID50 of B.1.1.529/Omicron” were not consistent with the main text. Please check your results.
9) For Figures 1B and 1C, even though the body weights of mice before the challenge were used to establish a baseline, it would be better to include mock-infected mice here as controls. Besides, were Figures 1D and 1E used the same mice of Figures 1B and 1C?
10) Line 153, in hamster studies, it seemed that the study used a lower dose of viral stocks for Omicrons BA.1 and BA.5 of 6.5 × 105 TCID50/animal when compared to 9 × 106 TCID50/Omicron-infected K18 mice?
11) Line 155-156, and Figure 2B, hamster studies, why only measured the body weight in the first four days post-SARS-CoV-2 variants infection? Four days post-infection might not show much weight loss, especially for WA1 and Delta strains. Besides, even though the body weights before the challenge were used to establish a baseline, it would be better to include mock-infected hamsters here as controls. In addition, in Figure 2A, the full names of the abbreviations should be provided in the figure legend.
12) Line 162, here for Omicron BA.1 or BA.5 strains, the study used a dose of 6.5 × 105 TCID50/animal or 5 × 103 TCID50/animal?
13) Figure 2B, Line 177-178, didn’t see the data of Body weight of “(and 5 dpi for WA1).”
14) Figure 2C, Line 179, the description here of 5 × 101 TCID50 of B.1.617.2 (Delta, 179 n=7) is not consistent with the main text. Please check your results. Besides, no TCID50 data from the lungs of hamster studies since it might be more relevant to viral replication?
15) Line 167, “the highest viral burden in lungs was measured on Day 3”. However, it seems like the highest viral burden in the lungs peaked on day 2, according to Figure 2D.
16) Lines 152-153, the doses here were not consistent with the doses in Lines 368-369.
17) Line 187, the description here, “9 x 106” was not consistent with the legend of Figure 3. Please check your results.
18) Line 188, the description here, “9 x 106” was not consistent with the legend of Figure 3. Please check your results.
19) Line 190, the description here, “9 x 106” was not consistent with the legend of Figure 3. Please check your results.
20) The dose data In Figure 3B wasn’t consistent with the corresponding Figure legend. Please check your results.
21) Line 210 and Line 215, “C57BL\6J” should be replaced by “C57BL/6”.
22) Figure 3E, K18 group, “G7” mice used the dose of “9 x 106”?
23) Line 198, the dose here “9 x 106 TCID50” for WT mice was not consistent with Figure 3E. Please check your results.
24) In Figure 3E, the viral load here by TCID50 assay was the same samples of Figure 3D (lung samples, B.1.1.529 SARS-CoV-2 after 3 dpi)?
25) Line 196, “1 × 104 TCID50/mg in the lungs”. However, it seems like “1 × 104 viral genome copy numbers/mg in the lungs” would be more appropriate here, according to Figure 3D.
26) Figure 3D and 3E, why the study checked the viral copy number and TCID50 in the lungs on Day 3 rather than Day 5-7 since the study showed that mice body weight loss peaked at day 7? On Days 5-7, the viral copy number and TCID50 might be higher.
27) Was Figure 3F used the same mice in Figure 3B and 3C?
28) The third Figure of Figure 4A, “Group 7 Omicron 9E6” means “K18-ACE2 Omicron 9E6”? And “Group 3 Omicron 9E6” means “C57BL/6 Omicron 9E6”?
29) Figure 5, Line 264, here “(C)” should be replaced by “(B).”
30) Figure 5, Line 266, here “(D)” should be replaced by “(C).”
31) In the last figure of Figure 5B, the study looked for the %CD8+ cells that GzB+IFNγ+TNFα+ all positive? If so, it was a different population from the third figure of Figure 4C(AIM+).
32) Suggests moving Figure 6 to supplementary files.
Author Response
Response to Reviewer 2 Comments
Major comments: 
In the manuscript, Toomer et al. found that different SARS-CoV-2 variants had different outcomes in mice and hamster models. Specifically, B.1.617.2 is more pathogenic in K18 mice while not in hamsters. B.1.1.529 showed an absence of clinical signs in both mice and hamster models, and it was able to infect wildtype C57BL/6J mice. While similar viral shedding profiles and titers between K18-hACE2 and 306 C57BL/6J mice were observed, B.1.1.529-infected K18-hACE2 mice had different T cell profiles compared to infected C57BL/6J mice.
The study is beneficial in the SARS-CoV-2 experimental Design and our understanding of COVID-19 animal models that are helpful in developing antivirals, vaccines, and other therapeutic compounds against COVID-19.
The study can be further improved by including the following suggestions listed in specific comments.
Specific comments:
Point 1. Title, “Characterization of Three Variants of SARS-CoV-2 in vivo Shows Host-Dependent Pathogenicity in Hamsters”, why were only hamsters mentioned here?
Response 1. We thank reviewer 2 for his/her thorough review. We intended to highlight that the hamster model showed host-dependent pathogenicity, while in mouse there is a more specific SARS-CoV-2 variant-dependent pathogenicity. Reviewer 2 is right, and we agreed to modify the title “Characterization of Three Variants of SARS-CoV-2 in vivo Shows Host-Dependent Pathogenicity in Hamsters, while not in K18-hACE2 mice”,
Point 2. Line 22, suggests replacing “ACE2” with “K18-hACE2”.
Response 2. We thank reviewer 2 for this correction, ACE2 was replaced with the proper full nomenclature K18-hACE2 to maintain consistency. Lines 20, 22 and 24 were corrected in the revised version of this manuscript.
Point 3. Line 19, suggests replacing “mice” with “K18-hACE2 mice”.
Response 3. We thank reviewer 2 for this correction, the suggested change was made.
Point 4. Line 21, “B.1.1.529-infected ACE2 mice had different T cell profiles compared to infected K18-hACE2 mice”, here, it should be “B.1.1.529-infected C57BL/6J mice had different T cell profiles compared to infected K18-hACE2 mice”.
Response 4. We thank reviewer 2 for the correction. The suggestion was incorporated in the revised version. We apologize for the multiple typos in this sentence.
Point 5. Introduction part, a brief introduction/summary should be provided about the SARS-CoV-2 animal models of mice and hamsters, and what questions the study will address.
Response 5. We agreed with reviewer 2 and the following paragraph was added in the introduction.
>>Animal models are required in SARS-CoV-2 preclinical research. In vitro, ex vivo and organoid models are key to revealing some molecular mechanisms of SARS-CoV-2 infection, animal models recapitulate the clinical and pathological characteristics of COVID-19 in humans are required to study viral pathogenesis, transmission, evasion strategies, disease etiology, host responses, therapeutic agents, and vaccines. Several animal models have been used during this pandemic. However, the literature is overwhelming regarding rodent models due to their low cost, convenient husbandry requirements, and ease of availability. However, the drawbacks of using mouse models for human viruses are species tropism, species specificity and immune response factors. Compared with other lab animals, mice offer many practical advantages, including small sizes, multiple well-established strains, clear genetic background, highly available research tools, and ease of genetic manipulation. Due to the low affinity of mouse ACE2 (mACE2) for S protein of SARS-CoV-2, mice cannot be efficiently infected with the ancestral wild-type SARS-CoV-2 variants. Therefore, the use of the K18-hACE2 transgenic mice expresses human ACE2, the receptor used by SARS-CoV-2 to gain cellular entry. The human keratin 18 promoter directs expression to epithelia, specifically the airway epithelia where the viral infection typically begins. Making K18-hACE2 susceptible to SARS-CoV-2 therefore useful for studying antiviral therapies against COVID-19. Syrian hamsters (Mesocricetus auratus), rapidly developed into a popular model as they naturally express ACE2 residues that recognize the SARS-CoV-2 spike protein making them susceptible to SARS-CoV-2 infection and recapitulating many characteristic features as seen in patients with a moderate, self-limiting course of the disease such as specific patterns of respiratory tract inflammation, vascular endothelialitis, and age dependence mimicking transmission and different courses of the wide spectrum of COVID-19 manifestations in humans.
Point 6. Figure 1A, the full names of the abbreviations should be provided in the figure legend.
Response 6. In the revised version, abbreviations are provided. Body weight loss (BWL) was changed to a more consistent body weight change (BWC).
Point 7. Line 108-109, “Animals infected with a dose of 1 × 103 TCID50 Delta/animal presented 100% mortality by Day 8 (data not shown).” How many animals did the study use? Suggests including this result in supplementary data for the comparison of the same dose of WA1/2020.
Response 7. We thank reviewer 2 for his/her suggestion. A supplementary Figure 1 with the summary of the LD50-Delta infected K18 hACE2 mice was included in this version.
Supplemental Figure 1. Survival in K18-human angiotensin-converting enzyme 2 (hACE2) transgenic mice. (A) Experimental Design in mice (n = 6) infected intranasally with four (4) 10-fold serial dilutions of B.1.617.2/Delta SARS-CoV-2. Body weight (B) and survival (C) were monitored daily for 10 days.
Point 8. Line 122, Line 124, and Line 126, the descriptions here “9 × 109 TCID50 of B.1.1.529/Omicron” were not consistent with the main text. Please check your results.
Response 8. Good catch reviewer 2! We thank reviewer 2 for his/her careful review. Our stock titer for that Lot was 2.9 x 108 TCID50/ml (this is the titration in Vero E6 over expressing hACE2 and TMPRSS2 receptors aka VAT cells) and it was corrected in the table of Methods, therefore the correct targeted dose was 8.7 x 106 TCID50 in 30 µL. This is the volume we used per intranasal inoculation per animal. The dose is correct in the figure and is now corrected in the text.
Point 9. For Figures 1B and 1C, even though the body weights of mice before the challenge were used to establish a baseline, it would be better to include mock-infected mice here as controls. Besides, were Figures 1D and 1E used the same mice of Figures 1B and 1C?
Response 9. We thank reviewer 2 for his/her suggestion. We do not have a group of non-infected or mock. For this type of studies, the difference between groups is more significant and does not justify the use of an additional group.
Yes, mice are from the same experiment. As the experimental design Figure 1A defines, we used 12 animals per group, however, survival indicates mortality in WA1 and Delta, animals that were found dead were not included further in body weight change and no necropsy was performed on them, therefore no tissues were collected. An interim sac was performed at 3 dpi in n=7 for WA1, n=7 for Delta and n=4 for Omicron. Consequently the number of animals of each group varies and was specified in the main text, material and methods and in the figure legend.
Point 10. Line 153, in hamster studies, it seemed that the study used a lower dose of viral stocks for Omicrons BA.1 and BA.5 of 6.5 × 105 TCID50/animal when compared to 9 × 106 TCID50/Omicron-infected K18 mice?
Response 10. We thank reviewer 2 for his/her suggestion. Yes, we wanted to target the same dose of BA1 which is straight viral stock (no dilution). However, our BA5 stock does not produce the same titer, in order to be able to compare the differences in the hamster model itself we decided to use the same dose of both omicron strains, which was the straight viral stock of BA5. Therefore BA1 need it to be diluted to 6.5E5 for both BA1 and BA5 to be able to compare among them.
Point 11. Line 155-156, and Figure 2B, hamster studies, why only measured the body weight in the first four days post-SARS-CoV-2 variants infection? Four days post-infection might not show much weight loss, especially for WA1 and Delta strains. Besides, even though the body weights before the challenge were used to establish a baseline, it would be better to include mock-infected hamsters here as controls. In addition, in Figure 2A, the full names of the abbreviations should be provided in the figure legend.
Response 11. We thank reviewer 2 for this observation. However, we do not have a group of non-infected or mock. For this type of study, the difference between groups is more significant and does not justify using an additional group. Abbreviations are provided in this revised version.
Point 12. Line 162, here for Omicron BA.1 or BA.5 strains, the study used a dose of 6.5 × 105 TCID50/animal or 5 × 103 TCID50/animal?
Response 12. We thank reviewer 2 for correcting this issue. Right, the correct dose for Omicron was added. “Shedding profiles from hamsters infected with 5 × 103 TCID50 of Delta, WA1, or 6.5 × 105 TCID50/animal of BA.1 or BA.5”.
Point 13. Figure 2B, Line 177-178, didn’t see the data of Body weight of “(and 5 dpi for WA1).”
Response 13. We thank reviewer 2 for this comment. The information is in Figure 2D. The reason is that at the beginning, when we established our first ID50 in hamsters with WA1 we did not know how long the studies will have significant data, so the WA1 studies were longer. We found that in hamster SARS-CoV-2 produces an acute infection that clears out after 5 days, therefore, we decided that 4 dpi was a cost-efficient window to get the proper readouts for other studies.
Point 14. Figure 2C, Line 179, the description here of 5 × 101 TCID50 of B.1.617.2 (Delta, 179 n=7) is not consistent with the main text. Please check your results. Besides, no TCID50 data from the lungs of hamster studies since it might be more relevant to viral replication?
Response 14. Thank you for the correction, the typo in the figure legend was corrected to 5 × 103 TCID50 of B.1.617.2 Delta.
Point 15. Line 167, “the highest viral burden in lungs was measured on Day 3”. However, it seems like the highest viral burden in the lungs peaked on day 2, according to Figure 2D.
Response 15. We thank reviewer 2 for this observation. Figure 2C shows viral burden by viral RNA copy number, figure 2D is shedding profiles, both can correlate or not, not all virus from lungs gets shed, we just can’t conclude that based on shedding profiles. The direct evidence that we have shown that Day 3 is the highest of what we measure. However, it might be that Day 2 has even higher titers, we just do not know.
Point 16. Lines 152-153, the doses here were not consistent with the doses in Lines 368-369.
Response 16. We thank reviewer 2 for this observation, the correction was made in this revised version.
Point 17. Line 187, the description here, “9 x 106” was not consistent with the legend of Figure 3. Please check your results.
Response 17. We thank reviewer 2 for this observation, the correction was made in this revised version 1 x 106 was changed to 9 x 106.
Point 18. Line 188, the description here, “9 x 106” was not consistent with the legend of Figure 3. Please check your results.
Response 18. We thank reviewer 2 for this observation, the correction was made in this revised version 1 x 106 was changed to 9 x 106.
Point 19. Line 190, the description here, “9 x 106” was not consistent with the legend of Figure 3. Please check your results.
Response 19. We thank reviewer 2 for this observation, the correction was made in this revised version 1 x 106 was changed to 9 x 106.
Point 20. The dose data In Figure 3B wasn’t consistent with the corresponding Figure legend. Please check your results.
Response 20. We thank reviewer 2 for this observation, the correction was made in this revised version 1 x 106 was changed to 9 x 106.
Point 21. Line 210 and Line 215, “C57BL\6J” should be replaced by “C57BL/6”.
Response 21. We thank reviewer 2 for his/her suggestion. We used and ordered C57BL\6J . The description is correct.
https://www.jax.org/news-and-insights/1989/july/profile-c57bl-6j
https://www.jax.org/news-and-insights/jax-blog/2016/june/there-is-no-such-thing-as-a-b6-mouse.
Point 22. Figure 3E, K18 group, “G7” mice used the dose of “9 x 106”?
Response 22. We thank reviewer 2 for this observation, the correction was made in this revised version.
Point 23. Line 198, the dose here “9 x 106 TCID50” for WT mice was not consistent with Figure 3E. Please check your results.
Response 23. We thank reviewer 2 for this observation, the correction was corrected in the Figure 3E in this revised version.
Point 24. In Figure 3E, the viral load here by TCID50 assay was the same samples of Figure 3D (lung samples, B.1.1.529 SARS-CoV-2 after 3 dpi)?
Response 24. We thank reviewer 2 for this observation. Yes, that is correct. For the K18 we performed TCID50 and PCR, but for the WT mice we only did TCID50 assay.
Point 25. Line 196, “1 × 104 TCID50/mg in the lungs”. However, it seems like “1 × 104 viral genome copy numbers/mg in the lungs” would be more appropriate here, according to Figure 3D.
Response 25. We thank reviewer 2 for his/her suggestion. Proper reference figure was inserted after the sentence to avoid confusion between Figure 3D and 3E.
Point 26. Figure 3D and 3E, why the study checked the viral copy number and TCID50 in the lungs on Day 3 rather than Day 5-7 since the study showed that mice body weight loss peaked at day 7? On Days 5-7, the viral copy number and TCID50 might be higher.
Response 26. We thank reviewer 2 for this observation. The rationale behind choosing Day 3 was our previous work with other variants however, this observation is valid, and it will be interesting to see if titers on Day 5-7 are higher in future studies.
Point 27. Was Figure 3F used the same mice in Figure 3B and 3C?
Response 27. Yes, those are the same animals described in the experimental design.
We apologize for the confusion and in this revised version explicitly describe it in the caption in Figure 3 [F] Virus shedding as viral genome copy number (by RT-qPCR) in oral swabs taken at 2, 3 and 4 dpi of K18-hACE2 and WT mice intranasally inoculated with 9 × 106 TCID50/animal (Groups 3 and 7 only).
Point 28. The third Figure of Figure 4A, “Group 7 Omicron 9E6” means “K18-ACE2 Omicron 9E6”? And “Group 3 Omicron 9E6” means “C57BL/6 Omicron 9E6”?
Response 28. We apologize for the confusion, yes “Group 7 Omicron 9E6” means “K18-ACE2 Omicron 9E6” in this revised version explicitly describes it in the caption and in the figure, we added WT and K18 to identify the differences.
Point 29. Figure 5, Line 264, here “(C)” should be replaced by “(B).”
Response 29. We thank reviewer 2 for this observation, the replacement was made.
Point 30. Figure 5, Line 266, here “(D)” should be replaced by “(C).”
Response 30. We thank reviewer 2 for this observation, the replacement was made.
Point 31. In the last figure of Figure 5B, the study looked for the %CD8+ cells that GzB+IFNγ+TNFα+ all positive? If so, it was a different population from the third figure of Figure 4C(AIM+).
Response 31. Yes, the far-right panel of Figure 5B shows GZB+IFNy+TNFa+ CD8+ percentages. This is different from the AIM+ 4C panel. The AIM+ 4C panel does not depict the triple positive population since both the control and treated animals had less than the unstimulated controls, and there was no significant difference between the control or treated animals so the data was considered negative. Instead, 4C shows the GzB single-positive data which was above the unstimulated controls.
Point 32. Suggests moving Figure 6 to supplementary files.
Response 32. We thank reviewer 2 for his/her suggestion. Figure 6 was moved to the supplementary SFigure2.
Round 2
Reviewer 2 Report
I think the authors have sufficiently addressed most of my concerns.
One more suggestion:
Supplemental Figure 1, Body weight should be Figure “(C)”, while survival should be Figure “(B)”. Besides, please double-check the Y-axis of Body weight loss.